# Deep Compressed Super-Resolution Imaging with DMD Alignment Error Correction

Miao Xu [1], Chao Wang [1], Haodong Shi [1], Qiang Fu [1], Yingchao Li [1], Lianqing Dong [2] and Huilin Jiang [1,*]

[1] School of Opto-Electronic Engineering, Changchun University of Science and Technology, Changchun 130022, China
[2] Beijing Institute of Space Mechanics & Electricity, Beijing 100094, China
[*] Correspondence: hljiang@cust.edu.cn

**Abstract:** In the field of compressed imaging, many attempts have been made to use the high-resolution digital micromirror array (DMD) in combination with low-resolution detectors to construct imaging systems by collecting low-resolution compressed data to reconstruct high-resolution images. However, the difficulty of achieving micrometer-level alignment between DMD devices and detectors has resulted in significant reconstruction errors. To address this issue, we proposed a joint input generative adversarial network with an error correction function that simulates the degradation of image quality due to alignment errors, designed an optical imaging system, and incorporated prior imaging system knowledge in the data generation process to improve the training efficiency and reconstruction performance. Our network achieved the ability to reconstruct $4\times$ high-resolution images with different alignment errors and performed outstanding reconstruction in real-world scenes. Compared to existing algorithms, our method had a higher peak signal-to-noise ratio (PSNR) and better visualization results, which demonstrates the feasibility of our approach.

**Keywords:** super-resolution; deep compressed sensing; alignment error correction

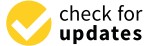



## 1. Introduction

With the development of single-pixel imaging technology, researchers have been looking for faster imaging speeds and larger pixel arrays. Therefore, scholars have proposed the theory of block-wise compressed imaging (BCI) [1]. This method can be seen as an extension of single-pixel imaging in plane space. Using high-resolution spatial light modulators to block-encode the scene in parallel, compressed data were sampled by low-resolution detectors and finally reconstructed into high-resolution images using a computationally efficient reconstruction algorithm. This achieved the acquisition of high-resolution images based on low-resolution detectors, which greatly reduced the cost of equipment and the amount of transmitted data. This could be applied to infrared detection [2], polarization imaging [3], and target tracking identification [4].

Due to their high resolution, simple control, and fast response, digital micromirror array (DMD) devices have been widely used as spatial light modulators in compressed imaging [5]. Although DMD devices have shown excellent performance in single-pixel imaging, they have also faced some challenges when used in BCI. First, due to the use of off-axis optics in imaging systems, they are susceptible to variations in the field of view, leading to distortions in various points of the image. Second, the manufacturing and calibration errors of the optical system, including aberrations in the lenses and misalignment between DMD and detector pixels, can all degrade the final image quality [6]. Finally, traditional algorithms for block reconstruction also have significant block effect issues.

To address these issues, various methods have been proposed. For example, the leakage of light from DMD and CCD pixels during registration and the effects of specific lenses on the super-resolution imaging results have been analyzed [7]. Mole patterns have

been loaded into the optical system, and calibration errors have been corrected based on Mole patterns [8]. Special coding schemes have been designed to reduce the effects of block effects [9]. Nonlinear mappings between measured values and the original object have been established using deep learning networks [10]. A joint input compression imaging network has been proposed, where customized coding modules are used to make the imaging degradation model input of the network [11]. However, traditional CS algorithms often result in slow reconstruction speeds and poor reconstruction quality. While innovative deep learning methods are seldom able to solve the problem of the actual optical system calibration errors that cause degradation.

In this work, we first analyzed the influence of optical system calibration errors on imaging to better simulate the degradation of imaging quality during optical imaging processes; then, a joint input generation adversarial network was proposed, which combined low-resolution degraded images and coding matrices with errors as network inputs to attempt to recover low-resolution images with unknown and complex degradation; finally, an experimental platform was built to validate the feasibility of this method in real-world scenarios, which successfully reconstructed high-resolution images with low sampling rates and reduced the error caused by misalignment.

## 2. Materials and Methods

### 2.1. Block-Wise Compressed Imaging System with DMD

As an extension of single-pixel imaging methods in the spatial domain, we designed a BCI system using DMD as a spatial light modulator (SLM). DMD is a reflective digital SLM that is composed of millions of micromirrors on the semiconductor silicon substrate [12]. Each micromirror is independently controlled and modulated by a tilting angle to modulate light. The block-based imaging system consists of five main components: the target, a distant objective, DMD, an imaging objective, and a visible light detector array. The imaging process is shown in Figure 1. Firstly, the telescope objective focuses the scene on the DMD; then, the resulting image is divided into multiple small blocks of the same size and is compressed by a common encoding matrix. Next, the modulated image is focused onto the detector array, where each pixel of the detector collects the intensity information of a target block. By varying the loaded patterns on the DMD, the detector collects a set of low-resolution compressed images. Finally, reconstruction algorithms are used to reconstruct high-resolution images. Compared to single-pixel imaging, BCI can achieve a higher resolution and use fewer sampling times.

In our system, each $4 \times 4$ micromirror on the DMD formed a large block and was projected onto one pixel of the detector. The formula for collecting data from each block is as follows:

$$y = \Phi x \tag{1}$$

In our system, $x(16 \times 1)$ is the original information corresponding to each block of the scene, $\Phi(M \times 16)$ is a measurement matrix, each row of $\Phi$ corresponds to a single pattern loaded on the DMD, $y(M \times 1)$ is the compressed sampling result, M is the sampling count of the detector and is also the number of patterns loading on the DMD. A parallel combination of the data collected from each block was performed, and the detector ultimately obtained M low-resolution images Y.

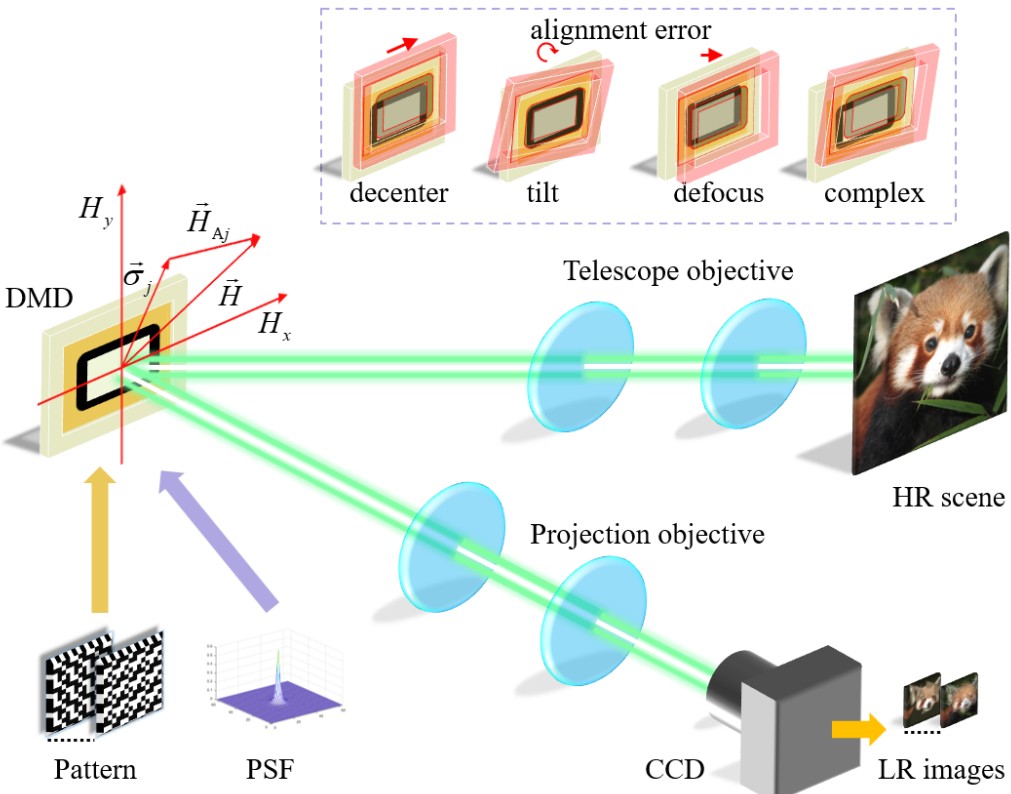

**Figure 1.** Block-wise compression imaging system diagram.

### 2.2. Effect of Alignment Error on Compressed Imaging

Due to the special structure of the BCI system, DMD devices need to achieve precise pixel-level alignment with the detector. However, it is difficult to achieve precise calibration during actual optical system construction. Alignment errors can bring aberrations to the actual optical system and have a significant impact on the reconstruction results. We needed to analyze the alignment errors and provide guidance for compensation in the reconstruction algorithm.

The classic wave aberration theory was used to analyze the impact of the alignment error on the imaging system. The alignment error caused the wavefront formed by the plane wave passing through the optical system to be no longer an ideal sphere. The actual wavefront was tangent to the ideal wavefront at the exit pupil, and the difference between the two wavefronts was called wavefront aberration. The off-axial optical system wave aberration of the DMD imaging system could be expressed using Seidel polynomials [13] as follows:

$$W(H, \rho, \phi) = \sum_{j} \sum_{p}^{\infty} \sum_{n}^{\infty} \sum_{m}^{\infty} (W_{klm})_j H^k \rho^l \cos^m \phi, \ k = 2p + m, \ l = 2n + m, \qquad (2)$$

In this equation, $W$ represents the optical path difference between the actual wavefront and the ideal wavefront at the pupil position, $W_{klm}$ the wave aberration coefficient, $k$, $l$, $p$, $n$, and $m$ represent the power series coefficients of each expansion, j represents the order number of each surface in the optical system, $H$ is the normalized field height, and $\phi$ is the angle between the field coordinates and the pupil coordinates.

When there is a decenter and tilt in the system, the contribution of the surface $j$ to the aberration of any field coordinate vector $\vec{H}$ in the viewing field should be calculated with

respect to the vector of aberration centers shifted by the vertex $\vec{\sigma}_j$ of the aberration, and the shifted vector, which is called the effective field of view $\vec{H}_{Aj}$.

$$\vec{H}_{Aj} = \left(\vec{H} - \vec{\sigma}_j\right), \tag{3}$$

From this, $\vec{H}_{Aj}$ is substituted into the wave aberration vector expression and yields:

$$
\begin{aligned}
W(H, \rho, \phi) = \quad & \sum_j (W_{020})_j (\vec{\rho} \cdot \vec{\rho}) + \sum_j (W_{111})_j \left[\left(\vec{H} - \vec{\sigma}\right) \cdot \vec{\rho}\right] + \sum_j (W_{200})_j \left[\left(\vec{H} - \vec{\sigma}\right) \cdot \left(\vec{H} - \vec{\sigma}\right)\right] \\
& + \sum_j (W_{040})_j (\vec{\rho} \cdot \vec{\rho})^2 + \sum_j (W_{131})_j \left[\left(\vec{H} - \vec{\sigma}\right) \cdot \vec{\rho}\right](\vec{\rho} \cdot \vec{\rho}) + \sum_j (W_{222})_j \left[\left(\vec{H} - \vec{\sigma}\right) \cdot \vec{\rho}\right]^2 \\
& + \sum_j (W_{220})_j \left[\left(\vec{H} - \vec{\sigma}\right) \cdot \left(\vec{H} - \vec{\sigma}\right)\right](\vec{\rho} \cdot \vec{\rho}) + \sum_j (W_{311})_j \left[\left(\vec{H} - \vec{\sigma}\right) \cdot \left(\vec{H} - \vec{\sigma}\right)\right]\left[\left(\vec{H} - \vec{\sigma}\right) \cdot \vec{\rho}\right] \\
& + \Delta \sum_j (W_{020})_j (\vec{\rho} \cdot \vec{\rho}) + \Delta \sum_j (W_{040})_j (\vec{\rho} \cdot \vec{\rho})^2,
\end{aligned}
\tag{4}
$$

The first-order characteristics include defocus ($W_{020}$), tilt ($W_{111}$) and translation ($W_{200}$), while the third-order characteristics include spherical aberration ($W_{040}$), coma ($W_{131}$), astigmatism ($W_{222}$), curvature ($W_{220}$), and distortion ($W_{311}$).

The generalized pupil function $P$ can be expressed as follows:

$$P = p \cdot \exp(iW), \tag{5}$$

where $p$ represents a transmittance function. The point spread function (PSF) of an optical system can be expressed as the generalized pupil function. Therefore, the relationship between *PSF* and wave aberration can be obtained as:

$$PSF = |FFT(P)|^2 = |FFT[p \cdot \exp(iW)]|^2, \tag{6}$$

where *FFT* represents Fourier Transformation. After encoding the image $Y$ with *PSF* convolution and adding noise $n$, the low-resolution image can finally be obtained. The degradation process can be expressed by the following equation:

$$Y' = Y \otimes PSF + n. \tag{7}$$

Due to the alignment errors of the DMD equipment in the BCI system mainly including defocus caused by the front and rear position deviations, decenter caused by up, down, left, and right deviations, and tilt caused by mounting angle deviations, it was necessary to focus on the first order characteristics.

### 2.3. Optical System Design

To analyze the impact of system calibration errors on the super-resolution imaging optical system of DMD, we designed an imaging optical system. This system consists of a telescope group and two relay lens groups. The DMD is placed in the second intermediate image plane. The main parameters of the optical system are shown in Table 1, and the design diagram is shown in Figure 2a. The modulation transfer function (MTF) of the optical system is shown in Figure 2b. It can be seen that when there was no alignment error, the optical system had good imaging quality, and the MTF value was close to the diffraction limit.

**Table 1.** Parameters of the optical system.

| Parameter | Value |
|---|---|
| Wavelength/nm | 390–780 |
| FOV(X/Y)/(◦) | 2.2°/2.2° |
| F/# | 1 |
| DMD array size/pixel | 1024 × 1024 |
| DMD pixel size/µm | 7.6 |
| Detector pixel size/µm | Detector pixel size/µm |
| 3.45 | 3.45 |
| Detector array size/pixel | 256 × 256 |

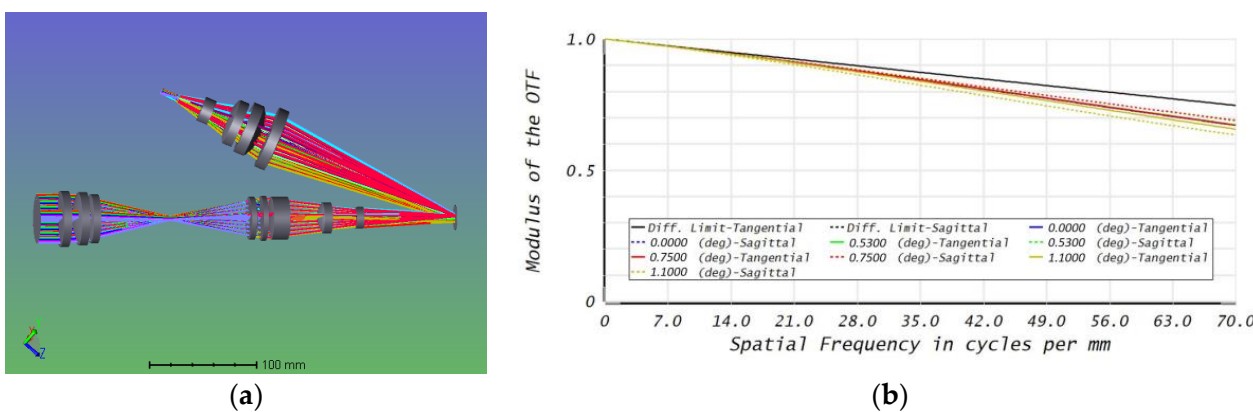

(**a**)  (**b**)

**Figure 2.** Schematic diagram of the DMD-based polarization optical system. (**a**) Structure of the optical system; (**b**) Modulation Transfer Function (MTF) of the optical system.

However, according to the analysis in the previous section, the optical system, when actually built, does not achieve an ideal imaging performance; therefore, we added alignment errors to the originally designed optical system, including the decenter, tilt, and defocus between the DMD and the detector. We obtained multiple PSFs with different error combinations, as shown in Figure 3. The position of the DMD in the actual system has undergone various deviations, resulting in a significant change in the PSF, which caused different degradation in the imaging quality.

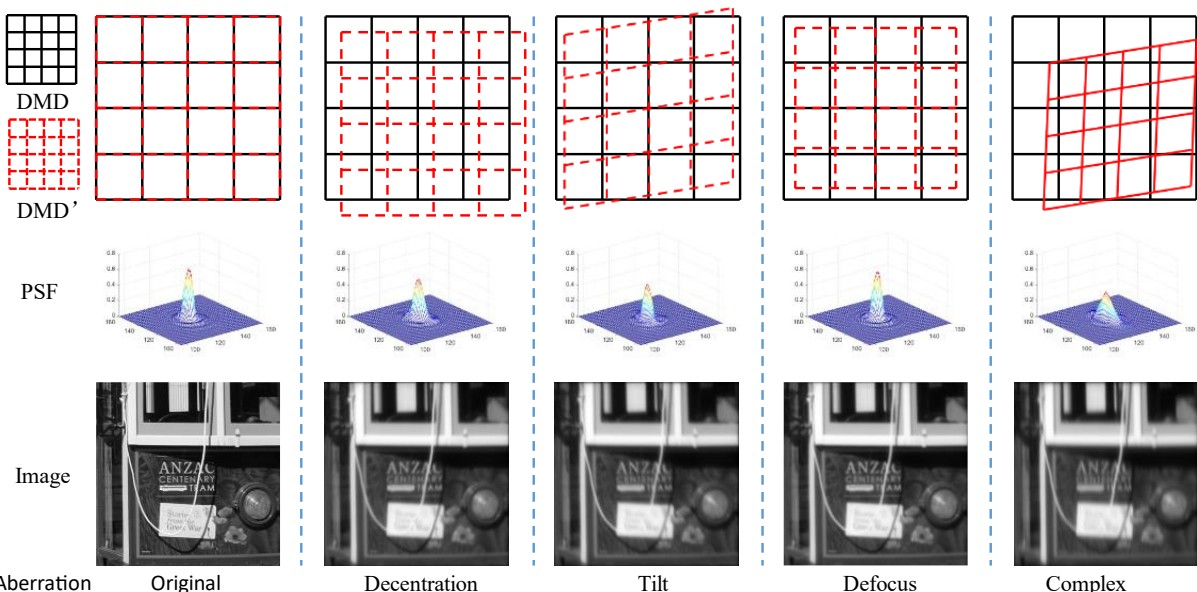

**Figure 3.** PSF and imaging results under different error conditions.

### 2.4. Super-Resolution Reconstruction Network

Different from the traditional compressed sensing image reconstruction algorithm, deep learning learns prior knowledge from the data, uses deep neural networks to establish a mapping relationship between the input and output, optimizes network parameters through large-scale data training, and inputs the sequence images collected by the detector into the compressed imaging system of the trained network, bypassing the complex computational process, resulting in a faster processing and the direct output of reconstructed high-resolution images [14].

Inspired by the Real-ESRGAN [15] network and the Joinput-CiNet network, we designed a compressed imaging super-resolution generational adversarial network based on error corrections. As shown in Figure 4, our network is mainly used to compensate for aberrations and optical alignment errors in optical imaging systems, achieving better reconstruction results. For this network, compressed encoding sampled low-resolution images with errors and high-resolution encoded images from which errors were input, and the reconstructed high-resolution images suppressed by error interference were output. Meanwhile, our network input multiple low-resolution images while maintaining the correlation among each block, effectively reducing the problem of the block effect in DMD compressed imaging and requiring no additional computational for the elimination of the block effect. We also used the U-Net as the discriminator for the network, which outputs the realness values for each pixel, and provides detailed per-pixel feedback to the generator.

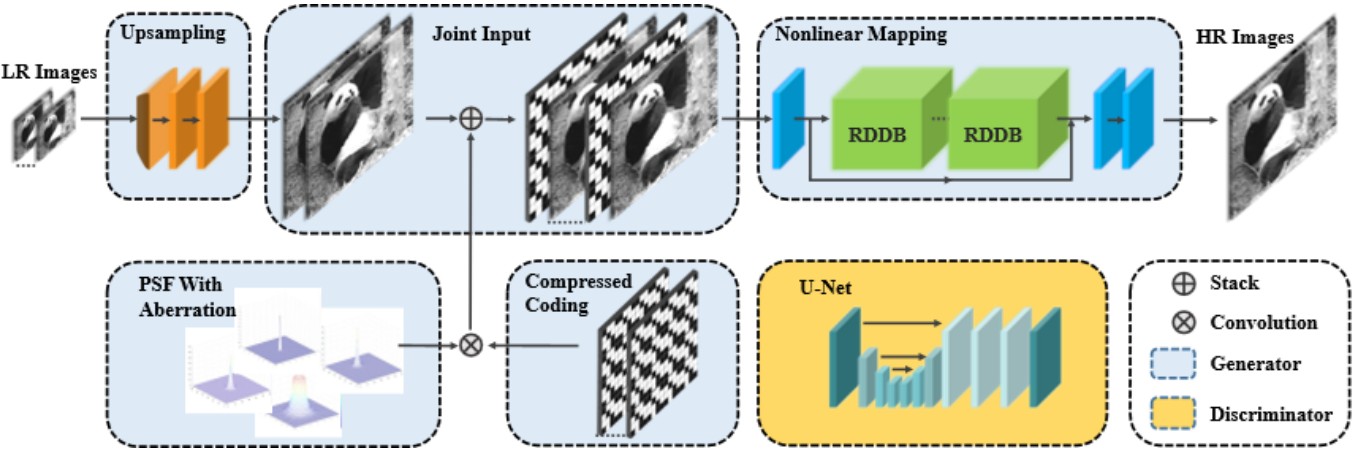

**Figure 4.** Flow chart of reconstructed network.

We improved the Real-ESRGAN network by using low-resolution compressed sampling data with alignment errors, DMD high-resolution encoded patterns, an optical system PSF with alignment errors as joint inputs to the network, and high-resolution images as the outputs. It has been verified that the combination of low-resolution images and compressed encoding, when input into the Joinput-CiNet network, can achieve better reconstruction results. Due to the four-fold difference between the low-resolution sampling and DMD size, it is difficult to train the network; therefore, we needed to align the two resolutions. We had to abandon the Joinput-CiNet network's PCA process to reduce the DMD encoding resolution and chose to enlarge low-resolution data to retain the full DMD encoding information and perform convolution with the encoding information and the PSF of the alignment mismatch to obtain the encoding matrix with errors. This approach improved the reconstruction results while simultaneously inputting the full encoding information and the alignment mismatch PSF.

### 2.5. Training Data

In order to train and evaluate the presented neural network, we used the DIV2K [8] and Flickr2K [9] datasets as high-resolution scene images. Data encoded sampling process

is shown in Figure 5. Using the previously mentioned degradation process, we compressed the high-resolution image into multiple low-resolution images and then convolved them with the PSF of the optical system designed. Finally, we calculated the low-resolution compressed image with alignment errors and simulated the process of DMD compression imaging through encoding. The 40 sets of PSF were calculated, which were then convolved with the high-resolution image randomly. We added noise and reduced the resolution of the convolved images; the resulting degraded image is shown in Figure 6. The 3000 sets of high-resolution images were used as the training data, 300 sets of images were used as the validation data, and 100 sets were used as the testing data. In addition, random horizontal and vertical flips were also selected during training. To increase the speed of training, the training HR patch size was set to 256. Similar to Real-ESRGAN, our net was trained with a combination of L1 loss, perceptual loss, and GAN loss, with weights (1, 1, 0.1), respectively.

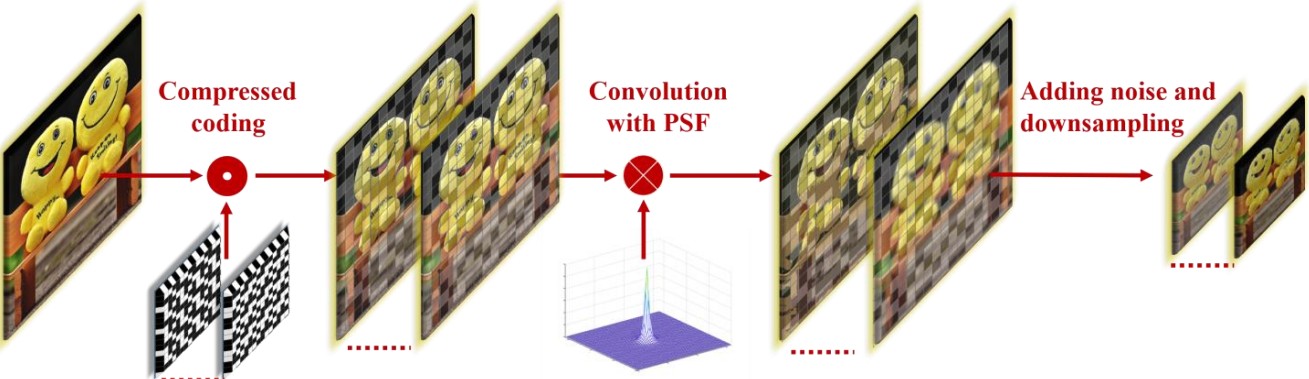

**Figure 5.** Data encoded sampling flow chart.

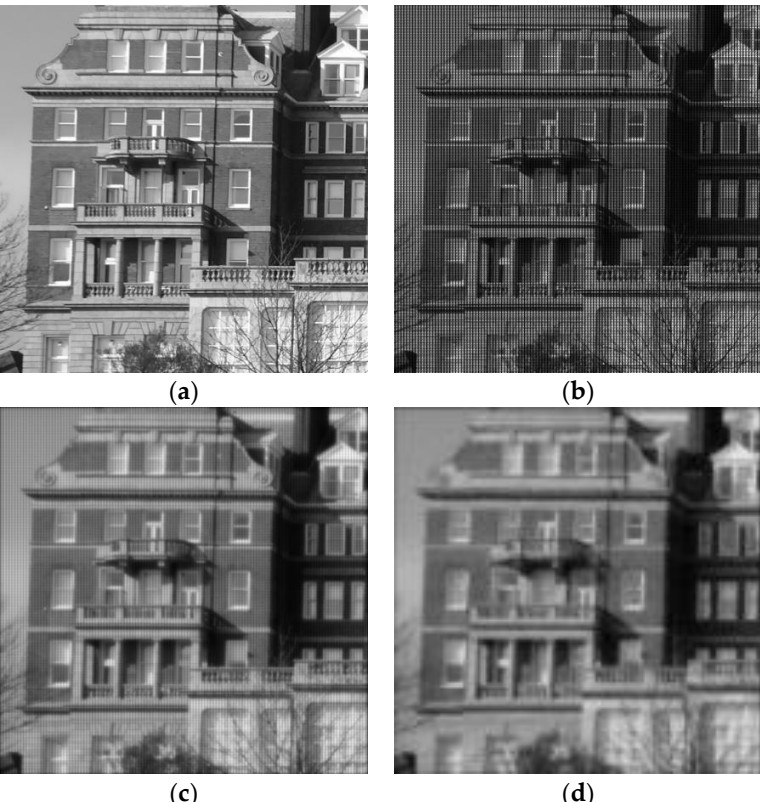

**Figure 6.** Effect of the degradation process of a high-resolution image. (**a**) Original image; (**b**) After DMD block encoding; (**c**) Image with alignment errors; (**d**) Image interpolated and amplified with noisy low-resolution detector.

## 3. Results

In order to apply this work to real scenarios, we conducted simulations and actual imaging experiments using our deep learning network to perform the super-resolution reconstruction of the collected images. The results from these experiments demonstrated the effectiveness of our approach.

### 3.1. Simulation

We divided the DMD into small blocks of size $4 \times 4$, with each block using the same 8-bit random Gaussian matrix as the sampling matrix. Then, we controlled the sampling rate by controlling the number of imaging times of the detector. Since the image magnification was $4\times$ and 16 images were acquired for complete sampling, the number of input low-resolution images corresponding to these three compression ratios was 1, 2, and 4. Super-resolution algorithms typically evaluate reconstruction results using PSNR and SSIM, and we calculated the values of the reconstructed images with different compression rates, as shown in Table 2. The reconstructed images with different compression rates of polarization are shown in Figure 7

**Table 2.** PSNR/SSIM values of reconstructed images with different compression rates of the test set.

| CS Rate | 1/16 | 1/8 | 1/4 |
|---|---|---|---|
| PSNR/SSIM | 23.3149/0.7649 | 26.9589/0.8846 | 28.0675/0.8723 |

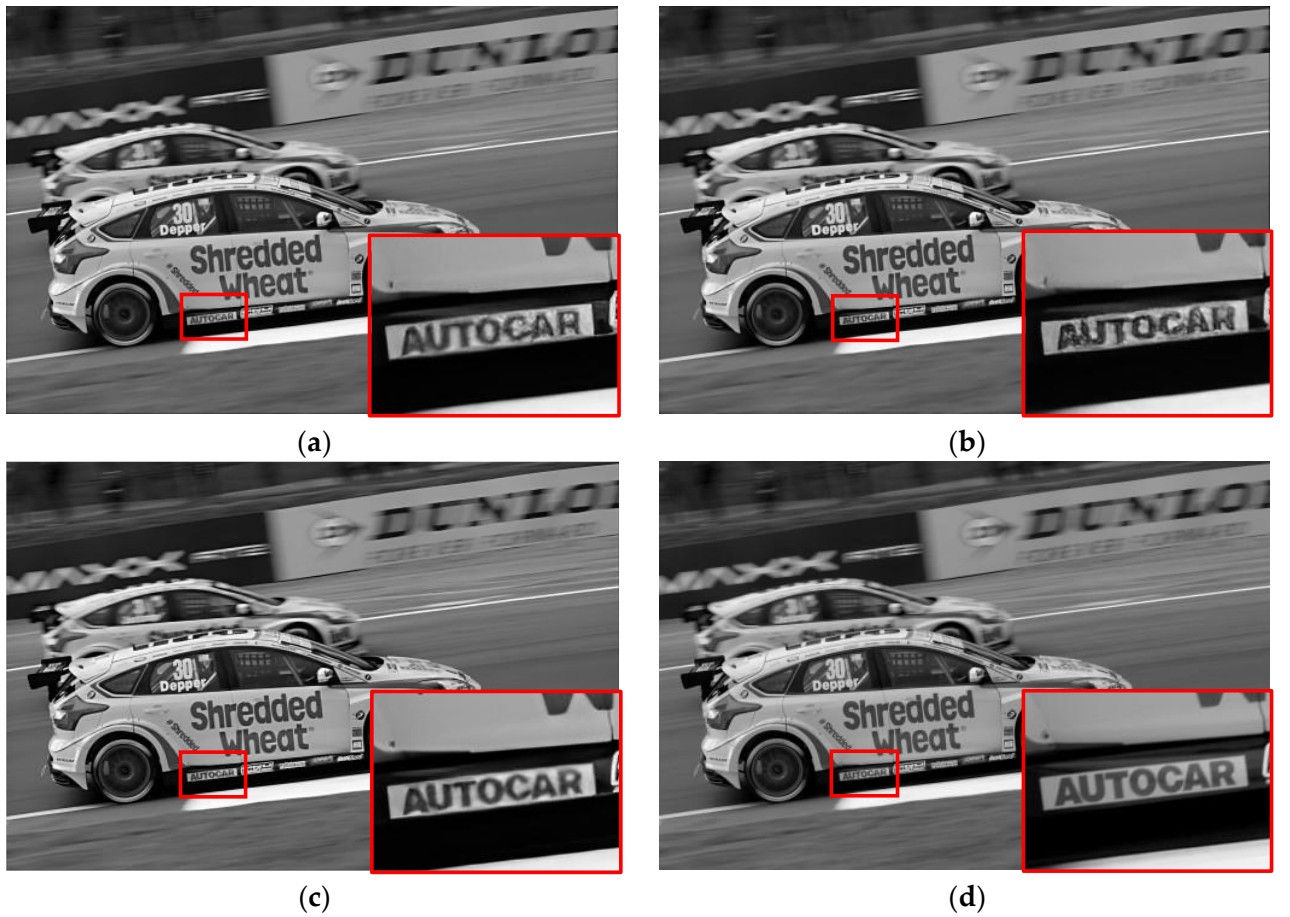

(**a**)       (**b**)

(**c**)       (**d**)

**Figure 7.** Reconstruction images with different sampling rates. (**a**) Sampling rates = 1/16; (**b**) Sampling rates = 1/8; (**c**) Sampling rates = 1/4; (**d**) Ground truth.

In order to evaluate the reconstruction effect of the algorithm, we compared our algorithm to the compressed sensing algorithm OMP and the deep learning networks Reconnet and Real-ESRGAN. The input of the OMP algorithm and Reconnet network on each block of data was obtained, directly outputting the reconstructed single block information and then combining each block of information to create a high-resolution image. Real ESRGAN is a single image super-resolution algorithm, with the input being a single low-resolution image that is directly collected without encoding, and the compression rate is equivalent to 0.0625. PSNR and SSIM are usually used to evaluate the reconstruction results. We evaluated the reconstruction effect of each algorithm under different alignment errors. The specific values of the reconstructed images under different alignment errors are shown in Table 3.

**Table 3.** The specific values of the reconstructed images under different alignment errors.

|  | Decenter | Tilt | Defocus | Complex |
|---|---|---|---|---|
| OMP | 12.6821/0.0462 | 12.6631/0.0448 | 12.6633/0.0475 | 12.6673/0.0434 |
| Reconnet | 21.4278/0.5843 | 21.2601/0.5772 | 21.6159/0.5902 | 20.8833/0.5720 |
| Real-ESRGAN | 18.6232/0.6923 | 18.3760/0.6733 | 18.8864/0.6923 | 17.7600/0.6604 |
| Ours | 24.6103/0.8305 | 23.6994/0.8108 | 23.6765/0.8201 | 23.4484/0.7983 |

In Figure 8, the reconstruction images of different algorithms at a sampling rate of 0.0625 and magnification of four are shown. It can be seen that the OMP and Reconnet network could not be well applied to data reconstruction with alignment errors, and the inevitable presence of mosaic artifacts was caused; therefore, further correction is necessary. The Real-ESRGAN network performed well in a single-image super-resolution, but its original degradation model and alignment error model still had significant differences, resulting in the loss of reconstruction details. Our algorithm could reconstruct images with good imaging quality. More details of the reconstructed image in different error cases are shown in Figure 9.

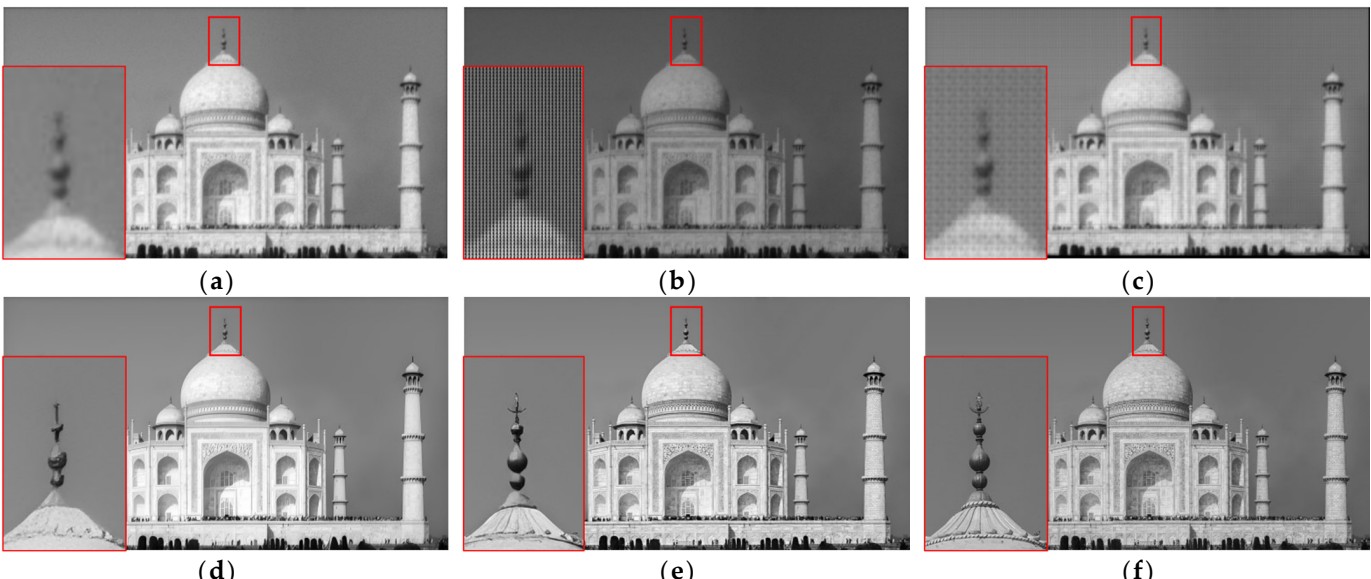

**Figure 8.** Comparison of the results of different super-resolution methods. (**a**) Bicubic; (**b**) OMP; (**c**) Reconnet; (**d**) Real-ESRGAN; (**e**) Ours; (**f**) Ground truth.

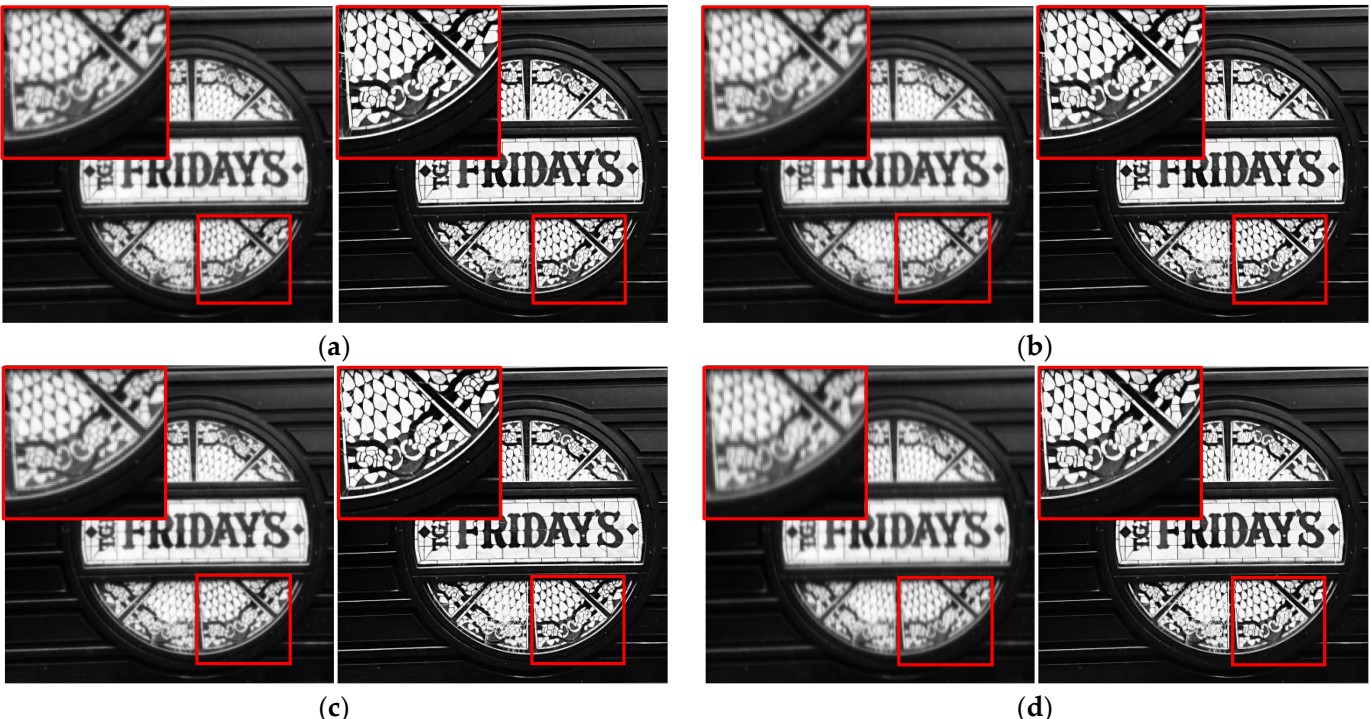

**Figure 9.** Low-resolution images with different errors and high-resolution reconstruction images. (**a**) Decenter (20 μm in both the X-axis and Y-axis directions); (**b**) Tilt (10′); (**c**) Defocus (20 μm); (**d**) Complex.

It can be seen that our algorithm exhibited excellent reconstruction effects at a low compression rate and dealt with different errors, both visually and numerically.

We used the Modulation Transfer Function (MTF) to analyze the reconstruction resolution. Our detector pixel size was 3.45 μm, and the corresponding system cutoff frequency was approximately 144 lp/mm. We tested black and white line pair images with different frequencies, added various alignment errors, and calculated the MTF of the reconstructed images, as shown in Figure 10. It can be seen that low-resolution images were difficult to visually distinguish at 25 lp/mm, while the images reconstructed by our algorithm still performed well at 50 lp/mm.

To evaluate the performance of our network, we tested the running time of OMP, ReconNet, Real ESRGAN, and ours in Figure 11. Reconstruction images with a size of (2040 × 1360) were used. The block size was (4 × 4). Our network included 16.6 million parameters for the generator and 4.37 million parameters for the discriminator. It can be seen that we performed well in the reconstruction effect, but there is still room for improvement in the running time. In future work, we hope to continue to optimize the reconstruction speed of the network.

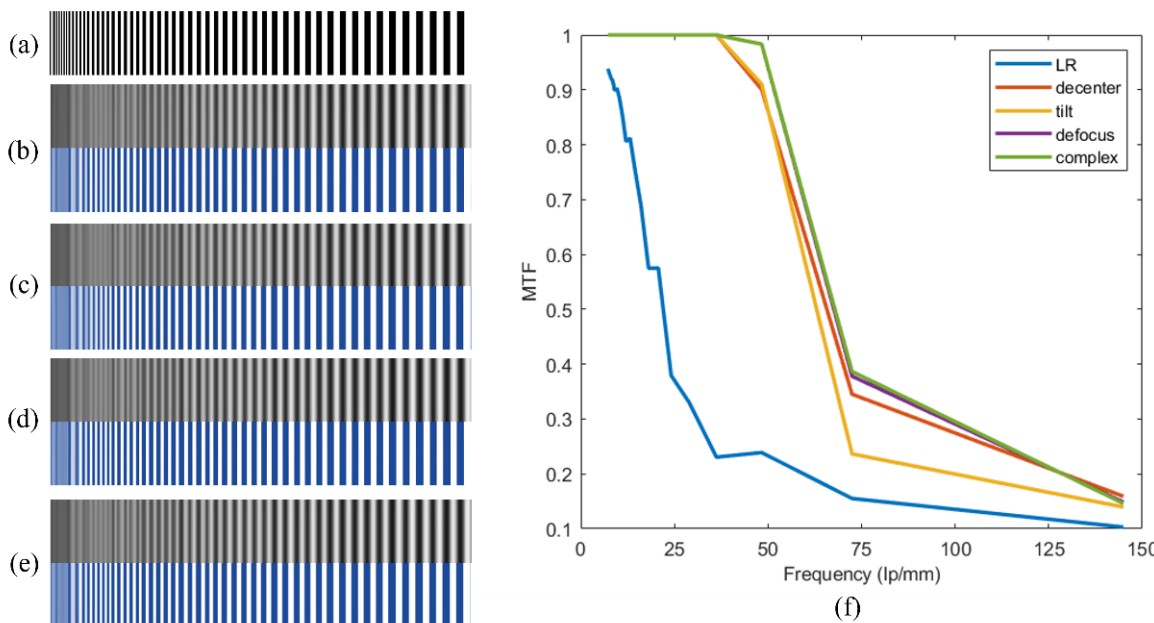

**Figure 10.** Low-resolution images and reconstruction results of black and white line pairs under different alignment error conditions. (**a**) Ground truth; (**b**–**e**) Low-resolution images with different errors and high-resolution reconstruction images; (**b**) Decenter; (**c**) Tilt; (**d**) Defocus; (**e**) Complex; (**f**) MTF images of reconstruction results with different alignment errors.

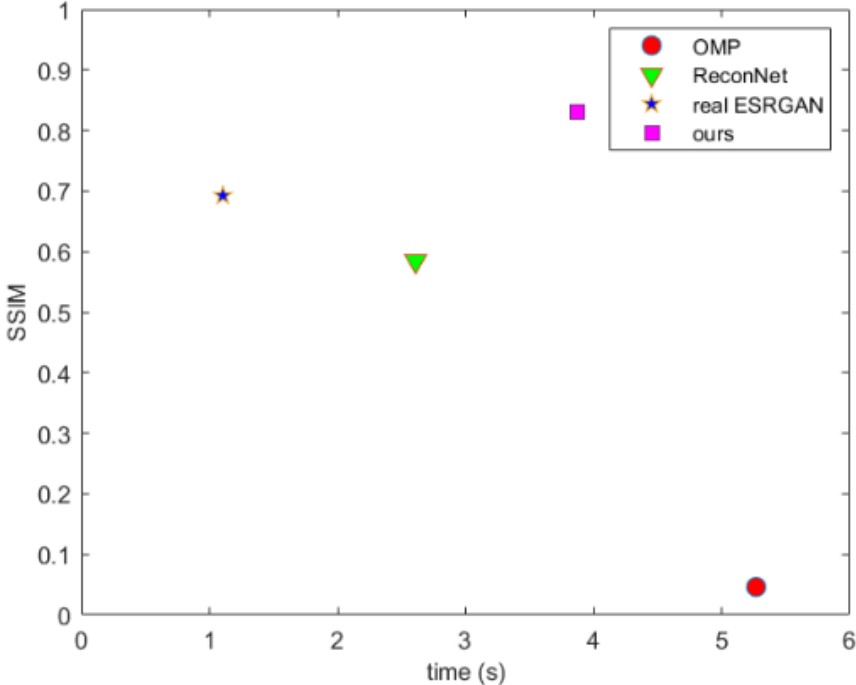

**Figure 11.** The runtime and SSIM of different reconstruction algorithms.

*3.2. Experiment*

To verify the application performance of our reconstruction network in a realistic scene, we selected core components such as TI Corporation's DMD, dual telecentric projection lenses, and LUCID Company's visible-light camera, and built a dual-arm reflective experimental setup, as shown in Figure 12a, the results of the experiment are shown in Figure 12b–f. Firstly, the scene was projected into the DMD by a convergent lens and a dual telecentric projection lens 1, with a resolution of 1920 × 1080. The size of each micro-mirror

in DMD was 10.8 μm, and the lenses could be switched at a high speed between ±12° directions. Next, the reflected direction of different micromirrors on the DMD was controlled to encode and modulate the scene. Due to the diagonal flip of the micromirror, we rotated the DMD 45° and the angle between the two arms at 24°. Finally, the encoded scene image was collected by the camera using the dual telecentric projection lens 2. The camera was also rotated 45°. We selected a region of 1024 × 1024 pixels on the DMD, corresponding to the 256 × 256 regions of the detector, for image acquisition. The entire device was placed on a flat optical plate to take photos of the outdoor scene.

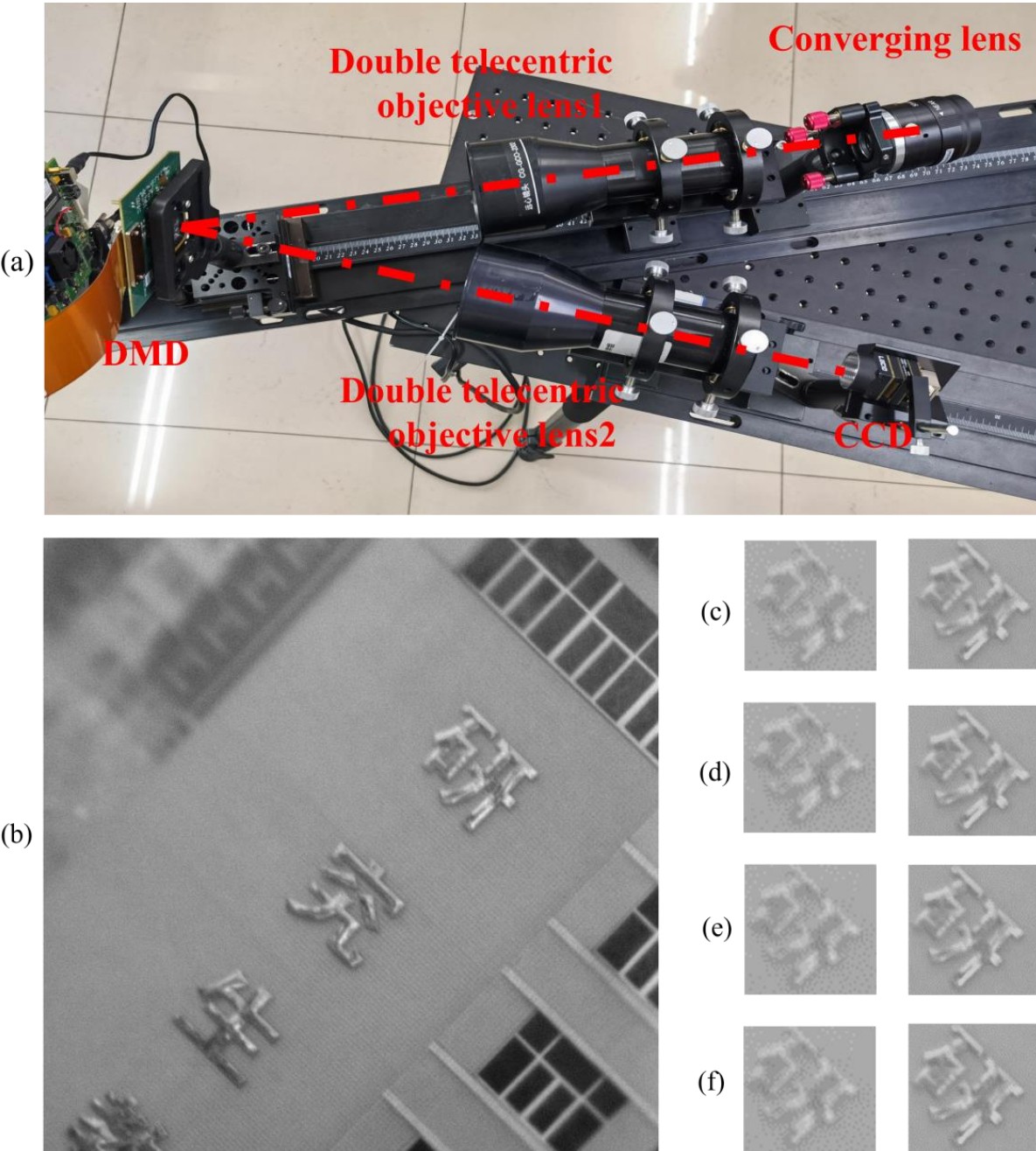

**Figure 12.** Experimental device and real imaging photography (**a**) Experimental device; (**b**) Reconstructed high-resolution image without alignment error; (**c**–**f**) Low-resolution images with different errors and high-resolution reconstruction images; (**c**) Decenter; (**d**) Tilt; (**e**) Defocus; (**f**) Complex.

Despite the large difference between the experimental data set and the actual installation error, we still improved the quality of the reconstructed image, indicating that our algorithm had certain robustness. This future improvement could be further improved by adding more real training data from actual photography.

## 4. Discussion

We built a deep learning network to achieve the super-resolution reconstruction of image compression sensing while compensating for the image degradation caused by alignment errors. During the reconstruction process, our network established a mapping relationship from low-resolution intensity images with alignment errors and compressed coding matrices to high-resolution intensity images. Compared with traditional methods, our deep compression and reconstruction algorithm not only had better reconstruction effects and faster reconstruction speeds but also the ability to correct the alignment error, making it more suitable for use in actual optical systems.

Currently, our method is limited by the resolution of DMD. In the visible wavelength range, our imaging method has not significantly improved its resolution. However, in the infrared detection dimension, by using appropriate training data sets, this resolution could be increased to approximately 2–4 times that of the original image. Our imaging method could use low-resolution infrared detectors to obtain high-resolution information, thereby reducing the cost of optical systems.

We designed a DMD compressed imaging optics system with Zemax optical design software and analyzed the effect of the alignment error on super-resolution reconstruction. The relationship between alignment error and the PSF can be explained with the wave aberration theory. We simulated the imaging degradation process caused by various alignment errors. When building the training data sets, we incorporated alignment errors into the optical system, generating a non-ideal PSF. When establishing the training dataset, we used our designed optical system to add alignment errors to generate nonideal PSF and used convolution to obtain images with errors, replacing the degradation model of common image algorithms. These methods are more suitable for real-world optics systems.

In order to better reconstruct the image, we took the compressed coding matrix as a priori knowledge and attempted to convolution the mismatch point spread function with the coding matrix to obtain the mismatch matrix. The mismatch matrix and low-resolution images were jointly input into the reconstruction network to improve the impact of optical system alignment errors on super-resolution imaging. The experiments have shown that our method significantly improved the reconstruction effect.

When the network trained from existing datasets was used to reconstruct images encoded and collected from real DMD optical systems, the reconstruction effect was already better than other traditional reconstruction algorithms, but the effect could still be further improved. Therefore, in future work, we look forward to collecting large amounts of data using DMD devices to meet specific demands, which will enable us to build a more accurate database for training.

## 5. Conclusions

In this article, we proposed a super-resolution compressed imaging method with an optical alignment error correction, which addressed the problem of image degradation caused by optical alignment errors in BCI. It took the encoded low-resolution image and high-resolution encoded matrix with optical alignment errors as joint inputs and reconstructed high-resolution images by GAN. Our method has been compared with existing methods, and it was found that the proposed method had a lower sampling rate, and the reconstructed images based on this method had more advantages in terms of evaluation indicators and visual effects. Additionally, our experimental setup demonstrated the potential of the imaging method and reconstruction network in future work, which will excite people's interest in applying deep learning and compressed sensing to practical imaging applications.

**Author Contributions:** Conceptualization, M.X. and C.W.; methodology, M.X. and C.W.; software, M.X.; validation, Q.F. and C.W.; data curation, H.S. and L.D.; writing—original draft preparation, M.X.; writing—review and editing, C.W.; project administration, Y.L.; funding acquisition, H.J. All authors have read and agreed to the published version of the manuscript.

**Funding:** This research was funded by [National Natural Science Foundation of China] 61805028, 61805027, 61890960, [Natural Science Foundation of Jilin Province] YDZJ202201ZYTS411, [Foundation strengthen domain technology fund] 2022-JCJQ-JJ-xx28, [Scientific and technological research projects of The Education Department of Jilin Province] JJKH20220742KJ, [Strategic Research Issues of Beijing Institute of Space Mechanics & Electricity], [Center of Space Exploration, Ministry of Education "Conceptual study on material exploration and in situ utilization of lunar underground lava tubes"] SKTC202101.

**Institutional Review Board Statement:** Not applicable.

**Informed Consent Statement:** Not applicable.

**Data Availability Statement:** Data underlying the results presented in this Letter are not publicly available at this time but may be obtained from the authors upon reasonable request.

**Conflicts of Interest:** The authors declare no conflict of interest.

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
