# Peer review of "Deep Compressed Super-Resolution Imaging with DMD Alignment Error Correction"

_photonics, doi:10.3390/photonics10050581_

Round 1

Reviewer 1 Report

The authors have proposed a joint input generative adversarial network with error correction function that simulates the degradation of image quality due to alignment errors, designed an optical imaging system, and incorporated imaging system prior knowledge in the data generation process to improve training efficiency and reconstruction performance. I still have the following suggestions for the authors to improve:

1.     Please make clear what kind of compressed sampling matrix is used and how to control the sampling rate?

2.     It is mentioned in the article that the alignment error is simulated by PSF, please clarify how to ensure that this error can be close to the real system error.

3.     The format of formulas in the text is not standard, and punctuation marks are missing after the formulas.

4.     Text formatting errors in lines 222-223.

5.     The format of the titles in Figures 7 to 9 is not standard.

It is suggest to polish the English writing.

Reviewer 2 Report

The manuscript focuses on the alignment problem between the DMD devices and the detectors, and introduces a joint input generative adversarial network to address this issue. Following there are some suggestions and questions for the authors to consider:

1.      In Fig. 1, all the physical qualities should be indicated in the content.

2.      In Fig. 2b, what’s the curves with different colors represent?

3.      In Fig. 4, it is not clear where the U-Net work.

4.      In Fig. 5, the last step is “adding noise” or “denoising”. If it is “adding noise”, then why it can help to obtain the final result.

5.      In the experiment result, more samples are suggested to verify the resolution improvement of the proposed method. And it should be indicated which kinds of main error are corrected.

6.      More references should be added to introduce the current related research status.

Minor editing of English language required.

Reviewer 3 Report

This paper discusses the deep compressed super-resolution imaging with DMD alignment error correction, which is suitable for publication in this journal. However, there are some minor suggestions for improving the clarity of the text:

1) In line 103, Eq. (2), it would be clearer to use k, l, m instead of p, n, m.

2) In line 115, for better understanding of the tip/tilt and decenter, the conjugation between the DMD and the object plane should be presented, possibly with a figure as in Table 1. Figure 1 does not clearly explain the concept.

3) In Fig. 9, lines 254-255, please specify the aberrations or errors in terms of numbers or percentages.

4) In Fig. 10, line 277, it appears that vignetting is present, suggesting that the experiment would benefit from proper illumination layout.

5) In lines 295-299, it is arguable that the method is better suited for the IR region. This may be because the training was done using visible datasets that have higher resolution than IR images.

6) Overall, it would be helpful to express the improvement in image enhancement in terms of MTF enhancement at a certain frequency

Reviewer 4 Report

The paper proposed a joint input generative adversarial network for compressed super-resolution imaging with DMD alignment error correction. It includes both simulation and real world scene. The proposed method achieved outstanding reconstruction in real-world scenes and demonstrates higher peak signal-to-noise ratio (PSNR) and better visualization results compared to OMP, Reconnect, and Real-ESRGAN.

Although, it is not clear if the novelty of the paper is design of the netwrok or using it for this applicaiton. Since it is claimed that the networked is designed by inspiration from Real-ESRGAN and Joinput-CiNet, I beleive that the network performance should be also evaluated. Therefore, comparisons like the number of parameters and the computational complexity of trained network in terms the number of operations or reconstruction time are needed.

Round 2

Reviewer 2 Report

The authors have improved the manuscript accordingly. It is suggested to be accepted.

It's ok.